Ecological solutions to reef degradation: optimizing coral reef restoration in the Caribbean and Western Atlantic

Lirman Diego dlirman@rsmas.miami.edu
Schopmeyer Stephanie
Department of Marine Biology and Ecology, University of Miami , Miami, FL , United States
Toonen Robert
Electronic publication date: 2016 Oct 20
Publication date: 2016
Volume: 4
Electronic Location ID: e2597
Received 2016 Aug 15; Accepted 2016 Sep 16
Copyright: © 2016 Lirman and Schopmeyer
Copyright year: 2016
Copyright holder: Lirman and Schopmeyer
License: This is an open access article distributed under the terms of the Creative Commons Attribution License, which permits unrestricted use, distribution, reproduction and adaptation in any medium and for any purpose provided that it is properly attributed. For attribution, the original author(s), title, publication source (PeerJ) and either DOI or URL of the article must be cited.
License URL: https://creativecommons.org/licenses/by/4.0/

Keywords: Coral gardening, Coral propagation, Coral reef restoration, Acropora, Threatened corals, Florida, Caribbean, Western Atlantic, Coral nurseries, Ecological services

Funding: NOAA, Counterpart International, The Nature Conservancy, MOTE Marine Lab, Florida’s Fish and Wildlife Commission, the US Army Corps of Engineers, and the National Science Foundation Funding for our restoration activities has been provided by NOAA, Counterpart International, The Nature Conservancy, MOTE Marine Lab, Florida’s Fish and Wildlife Commission, the US Army Corps of Engineers, and the National Science Foundation. The funders had no role in study design, data collection and analysis, decision to publish, or preparation of the manuscript.

==============================
Reef restoration activities have proliferated in response to the need to mitigate coral declines and recover lost reef structure, function, and ecosystem services. Here, we describe the recent shift from costly and complex engineering solutions to recover degraded reef structure to more economical and efficient ecological approaches that focus on recovering the living components of reef communities. We review the adoption and expansion of the coral gardening framework in the Caribbean and Western Atlantic where practitioners now grow and outplant 10,000’s of corals onto degraded reefs each year. We detail the steps for establishing a gardening program as well as long-term goals and direct and indirect benefits of this approach in our region. With a strong scientific basis, coral gardening activities now contribute significantly to reef and species recovery, provide important scientific, education, and outreach opportunities, and offer alternate livelihoods to local stakeholders. While challenges still remain, the transition from engineering to ecological solutions for reef degradation has opened the field of coral reef restoration to a wider audience poised to contribute to reef conservation and recovery in regions where coral losses and recruitment bottlenecks hinder natural recovery.

Introduction

The worldwide decline of coral reefs over the past several decades has been particularly devastating in the Caribbean where reefs have sustained massive losses, especially of reef-builders such as Acropora cervicornis, A. palmata, and Orbicella spp. (Gardner et al., 2003; Hoegh-Guldberg et al., 2007). These declines were driven and continue to be affected by disease and other stressors including the loss of the sea urchin Diadema antillarum, storm damage, and temperature anomalies (Aronson & Precht, 2001). The loss of reef-building taxa has contributed to decreases in reef structure and function, reef growth, fisheries habitat, coastal buffering, and biodiversity (Bruckner, 2002; Alvarez-Filip et al., 2009). The decline of key taxa has prompted conservation measures aimed at protecting remaining populations and accelerating recovery trajectories. These efforts in the Caribbean and Western Atlantic include: 1) the listing of taxa such as Acropora, Dendrogyra, and Orbicella as “threatened” under the US Endangered Species Act (National Marine Fisheries Service, 2006; National Marine Fisheries Service, 2014) and of O. annularis and O. faveolata as “endangered” and A. cervicornis and A. palmata as “critically endangered” in IUCN’s Red List of Threatened Species (2016); 2) the development of regional coral propagation and restoration programs (Young, Schopmeyer & Lirman, 2012); and 3) the drafting of species recovery plans for elkhorn (A. palmata) and staghorn (A. cervicornis) corals (National Marine Fisheries Service, 2015).

Engineering Reef Restoration

The field of coral reef restoration has grown considerably over past decades. Initially, restoration concentrated heavily on the design and execution of complex engineering projects aimed at quickly recovering or re-building the three-dimensional structure of damaged reefs impacted by physical disturbances, mainly ship groundings (reviewed in Precht, 2006). The main goal of these projects was to stabilize the reef framework and rehabilitate the lost structure that would take centuries to re-form without human intervention (Zimmer, 2006). The subsequent, ecological recovery of the damaged communities relied on a “build it and they will come” philosophy based on potential natural recruitment onto the newly deployed substrate (Kaufman, 2006). During some projects, soft and stony corals, either collected from the colonies surviving at the grounding site or harvested elsewhere, were added to the cement and limestone restoration structures after deployment, but large-scale ecological recovery was seldom realized. Examples of such expensive, large-scale projects include the restoration of the Maitland, Elpis, Houston, Wellwood, and Columbus Iselin ship-grounding sites in Florida (Wapnick & McCarthy, 2006) (Figs. 1A and 1B). However, due to their expense, complicated logistics, and permitting and legal considerations, these projects are commonly completed many years after the initial injury. For example, the initial injury to Looe Key Reef caused by the RV Columbus Iselin took place in August 1994 while the restoration of the damaged site was conducted in 1999 after a $3.76 million settlement in damage claims was reached. Unfortunately, the timing of such restoration projects coincided with the global decline of corals, thus limiting the likelihood of natural recovery of the original coral communities that were damaged in the first place. Damaged reef sites dominated by taxa like the now-threatened Acropora and Orbicella are especially problematic to restore as recruitment failure of these reef-building species prevents natural recovery (van Woesik, Scott & Aronson, 2014). In these cases, the coral community that develops on the restoration structures is often dominated by non-accreting macroalgae, octocorals, and sponges (Ruzicka et al., 2013), and “weedy” stony corals that are now dominant on degraded reefs (Green, Edmunds & Carpenter, 2008; Hughes et al., 2010). Assessments of the recovery trajectory of these engineering projects often found quick convergence to adjacent, undamaged coral communities, but only because these “control” communities had also undergone substantial declines and community shifts due to local and global stressors (Lirman & Miller, 2003). The technical difficulties associated with these projects resulted in significant resources spent on recovering relatively small areas and limit the global scope of these approaches. Finally, while these targeted approaches may work in response to specific needs such as the restoration of a portion of a reef, they are clearly inadequate for the recovery of threatened and endangered species or in response to large-scale ecological degradation.

Figure 1 Reef restoration structures.

(A) Cement modules used to restore the Maitland grounding site in Florida, (B) limestone boulders used to restore the Elpis grounding site in Florida, (C) nursery-grown A. cervicornis colonies attached to the modules used to restore the Wellwood grounding site in Florida (Photo credit: K. Nedimyer; Coral Restoration Foundation; http://www.coralrestoration.org/), (D) A. cervicornis colonies attached to ReefBalls in Antigua (http://www.reefball.org/), (E) A. cervicornis colonies attached to EcoReefs in Florida (http://www.ecoreefs.com/) (Photo credit: M. Johnson, The Nature Conservancy).

Ecological Reef Restoration

The limitations associated with just rehabilitating lost physical reef structure through engineering reef restoration projects created a demand for low-cost, low-tech approaches that could be implemented world-wide and focused on the ecological recovery of coral reefs. This recent emphasis turned the tables on prior engineering approaches while still retaining the ultimate goal of recovering an accreting, sustainable reef community that can provide the ecosystem services expected of a healthy reef by re-establishing the living components of the reef first and allowing reef accretion to proceed subsequently. The most widely used method for the ecological recovery of reefs is “coral gardening” (Fig. 2). This method, pioneered by Rinkevich (1995) and derived from terrestrial silviculture, is based on two tenets: 1) the collection and mariculture of coral fragments within nurseries; and 2) the outplanting of nursery-grown corals onto degraded reefs.

Figure 2 Coral gardening conceptual framework.

Conceptual model of the steps involved in the coral gardening framework, long-term goals, and benefits. The information in this model is based on our own research and activities, as well as information detailed in Johnson et al. (2011) and Rinkevich (2015).

Coral gardening differs from past ecological restoration projects in the Caribbean and Western Atlantic (Zimmer, 2006) and the Pacific (Jokiel et al., 2006) that relied on the transplantation of corals from a donor site to a damaged site (the “robbing Peter to pay Paul” approach) in that, during coral gardening, an initial small collection of corals is propagated within in situ or ex situ nurseries prior to outplanting onto degraded reefs (Fig. 3). The key to the success of coral gardening is, in fact, the nursery or grow-out stage where numerous techniques have been developed to maximize coral survivorship and productivity (Johnson et al., 2011). Because of enhanced survivorship and growth (achieved partly through pruning vigor; Lirman et al., 2010; Lirman et al., 2014), corals in the nursery can quickly provide a sustainable and expanding source of corals for ecological restoration, reducing the need for further collections from wild stocks that are severely degraded themselves. Limited initial collections, no sustained need for wild collections, high productivity while at the nursery, low cost relative to large engineering projects, and simple technical requirements have made coral gardening a preferred method for coral propagation and ecological reef restoration in the Caribbean and Western Atlantic (Young, Schopmeyer & Lirman, 2012), following similar trends from around the world (Rinkevich, 2014). Coral gardening projects to propagate Acropora were pioneered in the 1990’s and 2000’s in Puerto Rico (Bowden-Kerby, 1999; Bowden-Kerby, 2001; Hernández-Delgado, Rosado & Sabat, 2001; Hernández-Delgado, 2004), while Acropora propagation was initiated in Florida in 2001 by K. Nedimyer, 2016, personal communication.

Figure 3 Corals propagated using coral gardening methods.

(A) Coral tree (Nedimyer, Gaines & Roach, 2011) used in Florida to propagate corals, (B) A. cervicornis fragment, (C) A. palmata fragment, (D) Pseudodiploria clivosa fragment, (E) Orbicella faveolata fragment.

To quantify the increasing interest in the field of coral reef restoration around the world, we conducted a literature search of peer-reviewed journals, book chapters, and symposium proceedings using the keywords “Coral Reef Restoration,” “Coral Restoration,” “Reef Restoration,” “Coral Propagation,” “Coral Gardening,” and “Coral Nurseries” in Web of Science (Thomson Reuters), ProQuest ASFA (Aquatic Sciences and Fisheries Abstracts), and Google Scholar databases since 1980. A total of 268 papers were identified, with a steady increase in the number of publications over time. The period prior to 2000 had 29 publications, 2001–2005 had 37, 2006–2010 had 99, 2011–2015 had 103. The most publications in a single year (36) were recorded in 2015. In addition to an increasing trend in the number of publications, the proportion of publications reporting on engineering compared to ecological reef restoration solutions has changed over time, showing a clear shift of emphasis in the field. Prior to 2000, 51% of the publications were on engineering reef restoration, compared to 35% in 2001–2005, 18% in 2006–2010, and only 4% in 2011–2015. The increase in the number of citations is also reflected in an increase in the number of projects and programs. Young, Schopmeyer & Lirman (2012) conducted a review of coral restoration and propagation projects in the Caribbean and found > 60 individual projects from 14 countries using the coral gardening approach. Six years after this initial review, > 150 programs in > 20 countries now use the gardening method. The gardening of Caribbean and Western Atlantic corals has now reached ecologically meaningful scales where 10,000s of corals are being grown within nurseries and outplanted onto degraded reefs each year.

Engineering solutions may still be needed in cases where the substrate remains unstable and thus inadequate for successful transplantation or natural or assisted coral recruitment. Even in these instances, the recent development of ecological methods to stabilize loose rubble by deploying reef sponges may replace the use of cement as a binding agent (Biggs, 2013). The proliferation of coral gardening programs provides the added opportunity to combine both engineering and ecological restoration approaches and add a large number of nursery-grown corals onto the rehabilitated substrate. For example, in the Florida Keys, nursery-grown corals are now added to the limestone structures deployed to recover the Wellwood ship grounding site at Molasses Reef (Fig. 1C). Additionally, a number of mixed approaches exist in the Caribbean where nursery-grown corals are attached onto artificial structures, including cement structures (Jaap & Morelock, 1996); Reef Balls (Fig. 1D), ceramic EcoReefs (Fig. 1E), and electrified metal grids (van Treeck & Schuhmacher, 1997; Goreau & Hilbertz, 2005).

In addition to approaches like coral gardening that use adult colonies or coral ramets, the field of reef restoration using coral larvae reared ex situ has shown promising results in the Caribbean (Petersen et al., 2006; Petersen et al., 2008; http://www.secore.org), following earlier successful outcomes in the Pacific (e.g., Guest et al., 2010; Guest et al., 2014; Nakamura et al., 2011; Baria et al., 2012). In Curaçao, larvae of A. palmata reared from field-collected gametes were raised successfully in the lab for > 2 years and also outplanted onto wild reefs where they spawned at the same time as wild colonies (Chamberland et al., 2015; Chamberland et al., 2016). Coral gardening programs in Florida and the Caribbean now provide exciting synergistic opportunities to combine sexual and asexual propagation as coral nurseries hold (within common gardens) a large number of coral genets and ramets that are being used for gamete collection and fertilization research, and active ecological restoration (http://www.secore.org). By supplementing gardening activities with restoration using coral larvae, a greater impact on genetic diversity can be achieved. Moreover, the gametes and larvae reared from nursery stocks can provide key resources to support novel research activities such as coral hardening and assisted evolution (Rinkevich, 2014; Van Oppen et al., 2015) (Fig. 2).

Candidate Species

Gardening activities around the world are focused primarily on branching coral taxa that, due to their morphology, growth rates, and life histories characterized by asexual propagation through fragmentation, are ideal candidates for this approach (Rinkevich, 2014). During the initial stages of development of the coral gardening methodology in the Caribbean and Western Atlantic, the focal species were the branching acroporids (Figs. 3A and 3C). This framework is being currently expanded to include massive and encrusting coral species that were initially avoided due to their slow growth. The use of coral microfragments, as well as the development of re-skinning propagation techniques, are now providing an expanding stock of diverse nursery-raised coral species for gardening activities (Forsman et al., 2015). Some of the taxa being propagated in situ now include Orbicella (Florida, Belize) (Fig. 3D) and Dendrogyra (Florida, Dominican Republic), both taxa recently added to the US ESA, as well as Pseudodiploria (Florida) (Fig. 3E) and others.

Indirect Benefits of Coral Gardening

In addition to supplying corals for restoration, coral gardening programs provide a range of secondary benefits to ecosystems and local economies (Fig. 2). Coral gardening projects in the Caribbean and Western Atlantic have: 1) contributed to the rapid creation of fish and invertebrate habitat on depleted reefs by building new Acropora thickets that would otherwise take decades to form (Carne & Kaufman, 2015; Nemeth et al., 2016); 2) created genetic and genotypic repositories that can be used to enhance local diversity and recover genets eradicated by pulsed disturbances (Schopmeyer et al., 2012); 3) improved the physical connectivity of depleted adult populations by creating new reproductive populations in areas with large spatial gaps between surviving colonies (thus improving the likelihood of successful sexual reproduction); 4) provided a sustainable source of corals for experimental research (e.g., Enochs et al., 2014; Towle, Enochs & Langdon, 2015); 5) contributed corals and coral gametes that are reared in aquaria and zoos around the world where the benefits of coral restoration are showcased to millions of visitors; and 6) provided unique volunteering opportunities for citizen scientists to participate on the restoration process alongside practitioners (e.g., Rescue A Reef Program, http://www.rescueareef.com/; Coral Restoration Foundation, http://www.coralrestoration.org/).

But perhaps the most important indirect benefit provided by gardening programs are economic services in the form of employment and enhanced tourism opportunities (Abelson et al., 2015; Rinkevich, 2015). An excellent example of the ecological and economic synergisms created by coral gardening is provided by the program developed by the Puntacana Ecological Foundation in the Dominican Republic (http://www.puntacana.org/) that was initiated by A. Bowden-Kerby and expanded by V. Galvan, J. Kheel, and D. Lirman that has been in place for > 10 years. The program has outplanted > 15,000 staghorn corals onto reefs where this species had been eradicated due to algal overgrowth, disease, pollution, and coastal development. This program has also enhanced the local economy by: 1) restoring reefs that have become preferred dive sites used by local operators and hotels; 2) developing a “Coral First Aid” PADI dive specialty course taught by local dive shops to tourists; and 3) training local fishermen to become “coral gardeners” by providing them SCUBA certifications and employment opportunities to guide ecotourism excursions to nurseries and restoration sites. Transitioning fishermen from harvesting to gardening has the added benefit of reducing the impacts of unsustainable fishing practices on the reefs being restored. It is estimated that each fishermen hired as a coral gardener keeps an estimated 12.5 lbs of parrotfish per day on the local reefs, further improving reef conditions (Galvan, 2016).

Recipe for Long-term Success of Gardening Activities

A key component for the sustained success of ecological reef restoration is to develop a framework for responsible coral gardening, which requires a process to adequately train coral gardeners and regulate entry into the field by capable practitioners. The establishment of new gardening programs in the Caribbean region is commonly preceded by a training workshop offered by local or international experts where participants receive guidance on all aspects of the gardening process and a pilot nursery is populated with initial coral collections (Fig. 2). It is important that these workshops are attended by representatives from all stakeholder groups involved in coral restoration to ensure consistency and good communication among partners. In most countries, coral gardening activities require government permission. For example, practitioners in the USA are required to secure permits from local, regional, and federal agencies in charge of overseeing the restoration activities. These permits have strict monitoring and reporting requirements that keep programs accountable. Oversight by the local government is crucial to prevent the misuse of reef resources and to ensure a level of consistency and quality control of gardening operations. Local nursery operators should work closely with permitting agencies to ensure that best practices are used and that monitoring requirements are sufficient to track program success.

Local ownership of coral nurseries is another key factor determining the success or failure of the coral gardening framework. The relatively low cost and limited initial knowledge required to establish coral nurseries has resulted in an increasing number of new nurseries being deployed. However, the number of start-up projects is higher than the number of successful programs that effectively complete the two tenets of the gardening approach (nursery deployment and coral outplanting). The loss of resources or interest after the initial stages of a new program has resulted in “orphan” nurseries where corals continue to grow but are not maintained or, worse, never outplanted. In these cases, the nursery platforms collapse resulting in mortality of threatened/endangered corals (Fig. 4A). Untended nurseries with dead corals foster negative perceptions about reef restoration. To limit these unfortunate events, it is crucial that gardening programs have strong ownership shared by local stakeholders with established links to the community. Successful, long-term gardening programs are the result of partnerships among academic institutions, NGOs, government agencies, private businesses, and local community volunteers (see case studies in Johnson et al., 2011). Such partnerships allow for the co-management of nursery programs similar to the successful co-management of local fisheries resources (Yap, 2000; Cinner et al., 2012) and allow for gardening activities to be linked directly to other management tools such as Marine Protected Areas (MPAs) and watershed protection.

Figure 4 Examples from coral gardening projects.

(A) Damaged frame collapsed on the bottom due to lack of maintenance and coral pruning in Honduras, (B) staghorn outplants showing clear genotype-specific responses to the 2014 thermal anomaly in Florida, from no bleaching to paling and complete bleaching, (C) nursery-grown A. cervicornis spawning in Laughing Bird, Belize (Photo credit = Annelise Hagan and Fragments of Hope; http://fragmentsofhope.org/).

An important aspect of responsible coral gardening is to ensure that the scale of the nursery is commensurate with the resources available. The relative ease and low cost of nursery construction, coupled with the high survivorship and fast growth of corals within nurseries (Lirman et al., 2014) can create a scenario in which nurseries hold many more corals than can be managed or possibly outplanted, thus overwhelming nursery capacity and resources. While the tendency in those cases may be to fragment corals and expand the nursery, it is important to realize that every coral growing in a nursery is there only temporarily and needs to be outplanted. Good planning and a suitable exit strategy are thus needed to avoid nurseries from becoming too large to manage.

Even the best-planned restoration programs can lose funding, momentum, or interest. Some of the factors that have contributed to this scenario include: 1) lack of sustained funding beyond the nursery stage; 2) turnover in dive shop and hotel ownership and personnel; 3) vandalism and physical damage to nursery resources; and 4) loss of local support. In these cases, a clear exit strategy is needed to prevent the proliferation of orphan nurseries. An exit strategy should be an explicit part of the planning process and should clearly identify the scenarios that would trigger the interruption of a project and the steps needed to terminate the restoration project responsibly. At a minimum, an adequate exit strategy would require the outplanting of all nursery corals onto suitable reef habitat and the removal of all nursery materials from the site to prevent these materials from damaging nearby reef resources. This common-sense approach would mitigate negative perceptions of coral gardening in the local community and allow for the re-initiation of future projects in the same area if resources become available or conditions improve.

Remaining Challenges

The first advances in coral propagation within nurseries and the outplanting of nursery-grown corals were achieved by trial-and-error. With the maturation and expansion of this field, a number of programs have developed strong, science-based methods now published in the peer-reviewed literature and as manuals available online (Edwards, 2010; Johnson et al., 2011) that can be used by researchers, managers, local stakeholders, and any new entrant into the field to develop new programs in a systematic and scientifically defensible way.

The nursery stages of the coral gardening methodology have been extremely successful in the Caribbean and Western Atlantic region, with large numbers of fragments (> 50,000 kept in Florida nurseries alone), and an increasing number of species now routinely propagated. The next step, outplanting nursery-grown corals onto wild reefs, is still experiencing mixed results, with variable performance of outplants (Lirman et al., 2014). These challenges are clearly not simply logistical as numerous attachment methods, including nails, epoxy/cement, ropes, frames, and others, are being used successfully to secure outplants onto reefs (Johnson et al., 2011). Once outplanted, corals cement to the benthos and become natural components of the reef where they experience the same threats and challenges as wild corals. However, nursery-grown corals face novel challenges on present-day reef environments that differ from the ecosystems where they thrived decades ago. Corals are now commonly placed on reefs that have a significantly higher macroalgal cover and lower herbivore densities than historical levels. The loss of corals has potentially created a scenario in which coral predators (that have not declined to the same extent as corals) can target outplants and cause rapid mortality. This was observed in Florida where territorial damselfishes caused significant mortality to staghorn outplants soon after planting (Schopmeyer & Lirman, 2015) and in the Dominican Republic where the corallivorous fire worm Hermodice concentrate on newly deployed staghorn outplants (V. Galvan, 2016, unpublished data). Outplanted corals also face potentially detrimental water chemistry conditions where ocean acidification has created reef environments with low aragonite saturation states (Manzello et al., 2012; Manzello, 2015; Muehllehner et al., 2016).

An example of the challenges faced by outplanted corals was provided by the unprecedented, back-to-back bleaching events recorded in the Florida Keys in 2014 and 2015. During these events, both outplanted and wild colonies showed similar patterns of bleaching and mortality that were highly influenced by coral genotype and location (C. Drury, 2016, unpublished data) (Fig. 4B). Such disturbances provide set-backs in the restoration process but, if anything, highlight the need to continue to scale-up gardening activities and complement efforts with research to identify resistant coral holobionts, reef habitats, and combinations of Environments × Genotypes that can be used (or avoided) to build resilience and even mitigate the impacts of climate change (Rinkevich, 2014).

Considering the relatively young age of the gardening activities in the Caribbean and Western Atlantic, data are still lacking on the long-term survivorship of outplants. However, in Culebra, Puerto Rico, the site of the oldest gardening program in the Caribbean, staghorn outplants deployed in 2003 are still alive today (E. Hernandez, 2016, personal communication). Staghorn outplants have survived > 7 years in the Dominican Republic and outplants of both staghorn and elkhorn corals have survived > 6 years in Belize (L. Carne, 2016, personal communication). In Mexico, thriving first-generation elkhorn outplants are > 5 years old (G. Nava-Martinez, 2016, personal communication). In Florida, staghorn outplants have been shown to survive > 5 years, during which colonies have grown considerably, fragmented, and created new colonies. In addition, one extremely positive outcome of the gardening activities has been the observation of successful spawning of nursery and outplanted staghorn corals in Florida and the Caribbean (e.g., Dominican Republic, Belize; Fig. 4C). Moreover, elkhorn colonies reared from larvae were shown to spawn only 4 years after placement on reefs in Curaçao (Chamberland et al., 2016). The fact that nursery-grown corals (and corals raised from larvae) behave reproductively as wild corals lends support to using coral gardening to aid in the natural recovery of depleted populations.

The use of coral gardening methods for species and reef recovery are not without potential negative impacts that need to be considered. The two main ecological concerns, in our opinion, are disease impacts within nurseries and outplanted populations and genetic impacts on the extant populations. Diseases have been a major source of mortality to corals (particularly Acropora) in Florida and elsewhere (Aronson & Precht, 2001; Williams & Miller, 2005; Miller et al., 2014; Precht et al., 2016). By propagating a limited (but increasing) number of genets within densely populated nurseries, there is the potential that a rapidly progressing disease can decimate nursery stocks. An additional concern is the introduction or spread of a pathogen from nursery to wild reefs during outplanting. Unfortunately, limited data are available to evaluate the impacts of these concerns. However, reports from nursery practitioners indicate that the prevalence and impacts of diseases on staghorn coral are similar between nursery and wild populations, and between restored and wild populations (Miller et al., 2014) suggesting that environmental triggers, and not the gardening methods, are the main driver of disease prevalence. To limit the spread of disease within nurseries, practitioners commonly remove corals at the first sign of disease and nurseries often have a quarantine area removed from the main nursery where affected corals can be temporarily placed during outbreaks. While methods like excision of affected tissue and banding the diseased margin using epoxy have been tried to limit the spread and impacts of diseases, these interventions have not been especially successful (Miller et al., 2014). Nevertheless, no examples of complete nursery mortality have been reported and diseases commonly run their course leaving plenty of ramets unaffected to continue propagation. In Florida, only corals that are visually free of disease and appear in good health (normal coloration) can be outplanted onto wild reefs as per permit requirements, providing some level of protection. The increasing use of ex situ coral nurseries (e.g., Chamberland et al., 2015; Forsman et al., 2015) also raises the concern for the potential transmission of a disease vector from the lab to the field. While such cases have not been reported, practitioners in the US that want to outplant lab-reared corals are required to have their corals certified by a qualified veterinarian prior to transplantation. Targeted research is clearly needed to fully document the impacts of coral diseases within the gardening framework.

Another concern for gardening programs is the role that coral outplants can play on the genetic and genotypic diversity of wild populations, especially considering that the coral species being restored have experienced recent drastic bottlenecks in coral abundance. These concerns include the introduction of genotypes into novel environments where the fitness of a restored population may decline due to founder effects, genetic swamping, and inbreeding/outbreeding depression (Baums, 2008). In the last few years, genetic sampling has been routinely incorporated into nursery operations and the outcome of these studies can be used to support restoration activities and address these concerns. In Florida, recent findings of high genetic diversity within nursery stocks and wild reefs suggest that these concerns should be tempered and that local populations would benefit from the addition of new individuals (Drury et al., 2016). The incorporation of genetic sampling into nursery programs can be used to identify and target areas in need of active restoration. These target habitats would include areas low genetic or genotypic diversity that should be supplemented by nursery corals to increase resilience to local and climate impacts and the likelihood of successful fertilization, source reefs that supply larvae to connected reefs, and isolated reefs with low likelihood of sexual recruitment.

Conclusions

As reef restoration activities and programs in the Caribbean and Western Atlantic have transitioned from costly engineering projects into efficient ecological approaches, the coral gardening framework has “come of age” in the past decade and is now at the forefront of this important and emerging field. While challenges and data gaps remain, coral propagation and outplanting within a gardening framework conducted at meaningful scales and supported by strong science can play a significant role in the restoration of coral reef communities, the restitution of ecologic and economic services, and the recovery of threatened coral taxa. As the support for coral gardening grows, the next major step will be the documentation of benchmarks that can be used by practitioners to determine the efficacy of their efforts and impacts at the species and community levels.

While clearly beneficial for all regions of the world, coral gardening is especially important in the Caribbean and Western Atlantic where reef-building taxa are experiencing reproductive bottlenecks (Hughes & Tanner, 2000; Vermeij & Sandin, 2008; Williams, Miller & Kramer, 2008). Coral gardening is critical for the recovery of Caribbean species by providing a substantial source of large ramets that bypass the high-mortality of the early life stages of stony corals and are better able, due to their size and morphology, to survive algal completion and sedimentation once outplanted onto wild reefs (Rinkevich, 2005; Forsman, Rinkevich & Hunter, 2006).

Finally, it is important to temper expectations and note that no amount of coral gardening can fully recover a depleted species or ecosystem, especially when environmental and climate challenges remain. The goal of these activities should instead be to foster the natural recovery by re-establishing spatially connected populations with high genotypic diversity that can promote the successful sexual reproduction and natural recovery of the targeted species. Similarly, coral gardening and reef restoration cannot be the only tools employed. Reef restoration practitioners have recognized that local management tools such as watershed management, sustainable fishing practices, and the establishment of MPAs, among others, should be concurrently implemented to foster reef resilience.

We would like to thank all of our restoration partners and collaborators in Florida (NOAA, The Nature Conservancy, Coral Restoration Foundation, MOTE Marine Lab, NOVA Southeastern University, Fish and Wildlife Commission, Biscayne National Park) and the Caribbean (Counterpart International, PuntaCana Ecological Foundation, Central Caribbean Marine Institute, Fragments of Hope, Cape Eleuthera Institute) who have contributed to the development of a highly successful regional program for coral and species recovery based on the coral gardening framework. Field activities were supported by members of the Lirman Lab (J. Herlan, C. Drury, T. Thyberg, C. Hill, D. Hesley, K. Peebles, D. Burdeno) and volunteers of the Rescue A Reef program. Activities in the Dominican Republic were supported by A. Bowden-Kerby, V. Galvan, and J. Kheel. This manuscript was improved based on thoughtful reviews by R. Toonen, B. Shepard, and two anonymous referees.

Additional Information and Declarations

Competing Interests

Author Contributions

Data Deposition

The authors declare that they have no competing interests.

Diego Lirman conceived and designed the experiments, performed the experiments, analyzed the data, contributed reagents/materials/analysis tools, wrote the paper, prepared figures and/or tables, reviewed drafts of the paper.

Stephanie Schopmeyer conceived and designed the experiments, performed the experiments, analyzed the data, contributed reagents/materials/analysis tools, wrote the paper, prepared figures and/or tables, reviewed drafts of the paper.

The following information was supplied regarding data availability:

The research in this article did not generate, collect or analyse any raw data.

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
