# Peer review of "Ecological solutions to reef degradation: optimizing coral reef restoration in the Caribbean and Western Atlantic"

_PeerJ, doi:10.7717/peerj.2597_

## Round 0.1 · original submission · Minor Revisions

I now have three reviews back on your review paper, which is likely to become acceptable for publication following revisions. Two of the referees have a number of relatively minor suggestions to improve the manuscript, which should be simple to address. The third referee has the more substantive recommendation that the authors decide whether their primary focus is aimed at scientists or a rough guide for coastal managers on how to apply the coral gardening approach. In my own reading, I can certainly see where the referee is coming from and believe that the authors should give this criticism some serious thought in their effort to revise the manuscript.

·

Basic reporting

This is, overall, a good review of the current status of coral restoration, worthy of publication with a few revisions and additions. The authors focus on the Caribbean region, where the concept of “coral gardening” is having a number of positive impacts. The authors, however, do not mention an alternate restoration approach: utilizing annual coral spawning events to produce offspring for reef restoration. The gardening approach, as described in the article, is better suited to certain species (Acropora cervicornis, in particular), but is not equally suited to all species (e.g. Acropora palmata does not seem as adaptable to nurseries). A variety of parallel approaches is best- that is, there is no “one solution” to coral reef restoration. Therefore, in order for this to be a thorough review of the current state of ecological solutions to reef degradation in the Caribbean, there needs to at least be a brief summary of sexually-derived corals for use in restoration projects, noting the major accomplishments that have been made using sexually-derived corals for restoration, in particular in Curacao by scientists from CARMABI working in partnership with SECORE International.

The primary advantage of sexually-derived corals over the gardening approach using fragmentation is genetic diversity; gametes collected in the field and fertilized in the lab result in unique genetic combinations that were not present in the parental generation. This increased genetic diversity may produce lineages that are more resilient to global change. The typical gardening approach where only a few lineages of corals are grown in large quantities is analogous to a monoculture approach to farming, where “crops” may be more susceptible to disease, extreme weather events, and other natural impacts.

Sexually-derived corals are more expensive to produce [see Edwards, et al (2010) Evaluating Costs of Restoration, in Reef Rehabilitation Manual, p. 113-128]. However, technology and techniques are constantly being refined and there is much room to reduce overall costs, including the complete elimination of the nursery stage. This is one of the most costly and labor-intensive aspects of the coral gardening approach - as the authors acknowledge.

The authors do note, and show in figure 3, that one of the direct benefits of the nursery stage is enhanced sexual reproduction, perhaps aiding recruitment that has been negatively impacted by declining populations becoming more and more spatially disconnected, increasing recruitment failure, but I think there needs to be more attention paid to current research and restoration using sexually-derived corals.

Experimental design

As this is a review paper, there is no experimental design to review.

Validity of the findings

As this is a review paper, there are no findings - comments on the authors' discussion are summarized elsewhere.

Additional comments

Lines 220-221
The authors note that “successful, long-term gardening programs are the result of partnerships” - this is good. The success of the Coral Restoration Foundation nurseries in the Florida Keys, as well as the sexual reproduction work of SECORE has been supported by long-term partnerships with biologists working at many American zoos and public aquariums. Partnering with zoo/aquariums also has the benefit of reaching a wider audience through zoo and aquarium visitors - bringing the stories and successes of coral gardening to millions of people every year. It may be worth noting this as an aspect of public engagement and educational outreach about the topic of coral reef decline, aside from the brief mention in lines 170-171, which does not get at the depth of this partnership.

Lines 268-274
The discussions of mortality due to territorial damselfishes and Hermodice fire worms - the authors may note that caging coral recruits has been documented to increase survival. See Baria et al (2010) Journal of Experimental Marine Biology and Ecology 394: 149-153.

Lines 280-283
Here is an area where the authors note the need to continue to scale-up gardening activities and complement efforts with research “to identify combinations ... or genotypes ... to build resilience”. I think this is an instance where the authors should acknowledge how sexual reproduction techniques can also be used to complement the gardening efforts, and enhance gene diversity in the pool of corals in cultivation to mitigate climate change. Coral Restoration Foundation, along with partner organizations, are currently collecting spawn from their nursery corals. SECORE International, largely working in Curacao, but also conducting research in Mexico and Guam, and expanding into the Bahamas, is conducting restoration using coral recruits produced from spawn collected in the wild [see Chamberland, et. al. (2015) Restoration of critically endangered elkhorn coral populations using larvae reared from wild-caught gametes. Global Ecology and Conservation 4: 526-537].

In addition, laboratory-fertilized gametes collected from wild and/or nursery spawning events can provide scientists with the tools for building coral reef resilience through selective breeding and “assisted evolution”, which is an exciting new area of study [see van Oppen et al (2015) PNAS, 2307–2313, doi: 10.1073/pnas.1422301112]

Line 295
“...fact that nursery-grown corals behave reproductively…” - the same holds true for sexually-derived corals outplanted on the reef. Baria, Villanueva and Guest (2012) Bull Mar Sci 88:61-62 documented 3-year old Acropora millepora spawning with wild counterparts, and more recently, it was documented that Acropora palmata spawns 4 years after outplanting. See Chamberland VF, Petersen D, Latijnhouwers KRW, Snowden S, Mueller B, Vermeij MJA. Four-year-old Caribbean Acropora colonies reared from field-collected gametes are sexually mature. Bulletin of Marine Science, January 2016 DOI: 10.5343/bms.2015.1074

Lines 308-314
Great section highlighting the importance of this approach in the Caribbean, where reproductive bottlenecks are a major concern.

Lines 315-323
Another great section - yes - there needs to be a holistic approach to restoration, where gardening activities are complemented with fishery regulations/reforms, land management , MPAs, etc. There is no “one solution” to coral reef decline - all of these approaches have to be implemented in parallel.

Line 364 - there are 2 blue boxes in citation

Figure 3 - in “Novel Research” box, may include “Assisted Evolution” (see van Oppen et al, 2015 PNAS)

Reviewer 2 ·

Basic reporting

Please see general comments below

Experimental design

This is a review paper and there is no data analysis, therefore the submission is likely outside the scope of the journal

Validity of the findings

See general comments.

Additional comments

Review: Ecological solutions to reef degradation: optimizing coral reef restoration in the Caribbean and Western Atlantic

General comments:
The manuscript is a review of the rise of ecological restoration activities based on the ‘gardening concept’ which takes advantage of asexual propagation and a nursery phase for coral reef restoration. This brief review provides an overview of efforts primarily in the Caribbean, highlighting recent developments and shifts from engineering to gardening/ecological approaches, closing with a summary of challenges and some recommendations. The literature review is well written and a valuable overview, however it does not offer any empirical data or meta analysis. A keyword search is mentioned to highlight growth of the field, however the search was not graphed or contrasted with other terms (such as ‘artificial reef’) to illustrate shifts in focus in the literature. A flowchart diagram is presented intended to provide an overview of the nursery/restoration process, however there are several parts of the diagram that are dead ends or stem from nothing and the arrows should probably be re-thought. Overall I think the review is a valuable contribution and deserves to be published with minor revisions, although I am unclear if this type of review is the best fit for a journal such as Peer J, which tends to be more empirical (I can not comment on experimental design or robustness of data and statistics for example)… a closer look at the journal guidelines reveals the following; “PeerJ only considers Research Articles. It does not accept Literature Review Articles, Hypothesis Papers, Commentaries, Opinion Pieces, Case Studies, Case Reports etc. which may instead be submitted to PeerJ Preprints.”


Specific comments:

Line 38; Is this the most appropriate reference for documenting the decline of the Caribbean corals mentioned?
Line 43: the use of the word Keystone in ecology is related to the concept of an organism having an impact on the ecosystem disproportionate to it’s biomass (eg. a Keystone preditor). Although tempting to use this term for corals, I would instead say a key taxa..
Line 142; I think it would be useful to plot this data on a graph, although since publications as a whole are increasing rather rapidly, plotting just about any term will result in what appears to be a rapid proliferation unless that topic or method is no longer appropriate. A figure would still do a better job of conveying how the field is growing and it might be a worthwhile effort to compare the growth to a term such as ‘artificial reef’.

Line 182; putting overfishing first on this list makes it seem as though overfishing directly resulted in the species being eradicated, without making it clear that unmediated algal overgrowth was a more direct cause… this sentence could benefit from rewording to make cause/effect more clear or focus on synergistic causes.
Line 279; I think it should be emphasized that the nursery efforts may help to buy time and allow study of acclimation and adaptation, while raising awareness of climate change… I think calling this a ‘temporary set back’ is not really known and mitigating or reversing the impacts of climate change may be overstating the scale and effectiveness of restoration efforts… the gloomy science and projections need to be mentioned and acknowledged here as well.

Figure 3; the box Monitor corals, Develop Benchmarks is a dead end? Probably should be an arrow to Identify Outplant reef sites, which should come before coral outplanting… this section of the diagram needs a bit more thought to connect the arrows….. another example; I think Novel research should stem from Indirect benefits, whi

Reviewer 3 ·

Basic reporting

No Comments

Experimental design

No Comments

Validity of the findings

No Comments

Additional comments

The paper by Lirman and Shopmeyer provides an overview of the gardening approach to restoration as it has been applied to Caribbean and W Atlantic reefs. I found that the paper was generally clearly written and I found some useful information here. On the other hand, I was not quite sure if this is a review aimed at scientists or a rough guide for coastal managers on how to apply the coral gardening approach. I think the authors need to decide what they are trying to do here and either aim it at the broad reef science and conservation community and provide a really good overview of successes and failures to date using the coral gardening approach in the Caribbean with recommendations for future research; or, they need to write a research driven guide for managers on how to apply the gardening approach giving advice on what can go wrong and what is the current best practices (based on research). If they take the former approach I suggest providing much more quantitative information on success. For example, what increases in coral cover have been achieved due to restoration and how much has this cost? One last thing is that I felt there was not enough information about the potential flaws in the gardening method and it felt a bit like I was reading a funding proposal. There was hardly anything about population genetics for example and little information about actual costs of these approaches. Some additional minor comments below:

Line 112: “low cost” is listed as one of the advantages of coral gardening, but is it really low cost? Low compared to what? I don’t think any restoration efforts are really low in cost, but it could be argued that coral gardening approaches are much cheaper than efforts involving artificial reefs. Suggests saying “relatively low cost” or “low cost compared to engineering approaches”.

After line 118: contrast this with something about the disadvantages of the gardening approach.

Line 119-132: I’m not really sure if this literature search exercise is really that relevant. It is not that surprising that the number of papers on this topic has increased over time, what would be more interesting is to show that there has been increases in one area and deceases in others. For example has research interest in engineering type solutions deceased with time?

Line 320: “can’t” should be “cannot”.

Figure 3: This flow chart could be really good, but I found it missing important steps. For example, shouldn’t one start with the goals of restoration in the first place? E.g., are you trying to restore an endangered species, reef function, herbivory etc? This will guide the approach that you take. Also, I feel this could be improved if you used more of a decision tree type approach. For example, you start by asking the questions such as: what scale of restoration are you trying to achieve, do you have multiple species, do you have volunteers/local stakeholder interest, etc? Each of these leads to different pathways in the decision tree, some may lead you to use specific techniques, some may lead you to decide not to do restoration at all!

---

## Round 0.2 · accepted · Accept

Having read through your response to the referee comments and the reviews again, I see no reason to send this back out to review. I can see your point about the tone of the paper, and two of the referees were quite enthusiastic about that and already shared your opinion about it being useful as is. I feel that your additions have addressed the majority of the concerns of the reviewers and that this final point is more stylistic than academic, so I am happy to move your manuscript forward into production at this point.